# Ultrarapid Microwave-Assisted Synthesis of Fluorescent Silver Coordination Polymer Nanoparticles and Its Application in Detecting Alkaline Phosphatase Activity

**DOI:** 10.3390/molecules28041892

**Published:** 2023-02-16

**Authors:** Kanglin Pei, Di Li, Wenjing Qi, Di Wu

**Affiliations:** 1Chongqing Key Laboratory of Green Synthesis and Applications, College of Chemistry, Chongqing Normal University, Chongqing 401331, China; 2Department of Pharmacy, the Second Affiliated Hospital of Chongqing Medical University, Chongqing 401331, China

**Keywords:** fluorescent silver coordination polymer nanoparticles, alkaline phosphatase, microwave-assisted synthesis, ascorbic acid, ascorbic acid 2-phosphate

## Abstract

Fluorescent silver coordination polymer nanoparticles (Ag-TPA CPNs) were synthesized using a combination of terephthalic acid (TPA) and silver nitrate via an ultrarapid microwave-assisted strategy within 15 min. The Ag-TPA CPNs displayed a high fluorescent quantum yield (QY = 20.19%) and large Stokes shift (~200 nm), with two emission peaks at 490 nm and 520 nm under an excitation wavelength of 320 nm. A fluorescent “turn-off” method using fluorescent Ag-TPA CPNs was applied to detect the alkaline phosphatase (ALP) activity on the basis of the ALP-catalyzed hydrolysis of ascorbic acid 2-phosphate (AA2P) to ascorbic acid (AA), and the AA product triggered the reduction of Ag^+^ ions into silver nanoparticles. The fluorescent lifetime of Ag-TPA CPNs decreased from 3.93 ms to 3.80 ms after the addition of ALP, which suggests that this fluorescent “turn-off” detection of ALP activity is a dynamic quenching process. The fluorescent intensity had a linear relationship with the concentration of ALP in the range of 0.2–12 mU/mL (r = 0.991) and with a limit of detection (LOD) of 0.07 mU/mL. It showed high selectivity in ALP detection towards metal ions and amino acids, as well as other enzymes such as horseradish peroxidase, glucose oxidase, tyrosinase, trypsin, lysozyme, and superoxides. When it was applied for the fluorescent “turn-off” detection of ALP activity in serum samples, mean recovery levels ranging from 99.5% to 101.2% were obtained, with relative standard deviations (RSDs) lower than 4% accuracy. Therefore, it is an efficient and accurate tool for analyzing ALP levels in biosamples.

## 1. Introduction

Alkaline phosphatase (ALP) as a zinc-containing dimeric enzyme has the specific ability to remove phosphate groups from a variety of phosphorylated substrates, such as nucleic acids, proteins, and other phosphate-group-containing molecules (such as ascorbic acid 2-phosphate, AA2P) [1,2,3]. ALP is widely distributed in various tissues of living organisms. In biological systems, the ALP activity is closely associated with the health status of living organisms. ALP contained in serum is an important biomarker for several diseases, including liver dysfunction, leukemia, breast and prostatic cancers, diabetes, and bone and kidney diseases [4,5]. Therefore, it is of great significance for biomedical diagnoses to construct a simple, sensitive, and accurate biosensor to monitor ALP activity. Many approaches have been established for the measurement of ALP activity, including surface-enhanced Raman scattering [6], colorimetry [7,8,9], electrochemistry [10,11], chromatography [12], chemiluminescence [13], electrochemiluminescence [14], and fluorescence [5,15,16] strategies. Among these methods, fluorescent methods with high sensitivity, fast response times, and real-time detection capability have attracted particular interest [5,17,18]. Most fluorescent assays for ALP activity are based on the ability of ALP to remove phosphate groups from a variety of substrates containing phosphate groups and to cause fluorescent “turn-on” or “turn-off” changes [19]. Moreover, the reported fluorescent probes include small molecules [8], organic nanoparticles [4], infinite coordination polymers [5], gold nanoclusters [20], and metal–organic frameworks [21]. Sometimes these methods are limited by the complex synthesis of fluorescent probes or enzyme substrates, relatively poor photostability, toxicity of heavy metal elements, or small Stokes shifts. Therefore, it is of great significance to explore the use of fluorescent probes for the determination of ALP activity with simple and rapid synthesis processes, high quantum yields, and large Stokes shifts.

Coordination polymers are hybrid materials composed of metal ions and molecular bridging ligands [22]. They can be designed to synthesize a variety of materials with different sizes, structural characteristics, and physicochemical properties via the self-assembly of different metal ions and bridging ligands. Over the past decades, coordination polymers have been applied in various fields such as separation [23], chemical and biochemical sensing [24,25], drug delivery [26], cellular imaging [27], and heterogeneous catalysis [28]. Among these applications, coordination polymers have been widely utilized as fluorescent probes. There are two types of common metal ions that are used as metal nodes, namely lanthanide ions and transition metal ions [29]. For example, various lanthanide coordination polymers self-assembled from organic ligands and lanthanide ions have been designed to detect ALP activity. Xiao and his co-workers developed bimetallic lanthanide coordination polymer nanoparticles (Tb-GMP-Eu CPNs) for the sensing of ALP activity [30]. However, lanthanide ions are becoming increasingly valuable as rare earth metals, limiting their further application. Compared with lanthanide ions, transition metal ions are relatively cheap and easy to obtain. Thus, silver coordination polymers as transition metal elements have attracted the attention of researchers [31,32,33]. Most synthesis processes for silver coordination polymers are time-consuming, complex, or exhibit low quantum yields, which limits their application in biological analyses. Therefore, exploring facile and rapid methods to prepare silver coordination polymers with high fluorescent quantum yields or large Stokes shifts is extremely urgent for the development of the coordination polymers.

Generally, microwave-assisted synthesis, with the significant advantages of its super-rapid and higher yields, has exceeded traditional synthetic approaches. In particular, the synthesis of silver coordination polymers with high quantum yields and large Stokes shifts within several minutes using microwave irradiation is rarely reported. Inspired by these factors, terephthalic acid (TPA) has been employed as a bridging ligand and silver ions as metal nodes to construct fluorescent coordination polymer nanoparticles using a fast self-assembly strategy under microwave irradiation within 15 min. The as-prepared Ag-TPA CPNs display a high fluorescent quantum yield (QY = 20.19%) and large Stokes shift (∼200 nm) with two emission peaks at 490 nm and 520 nm under excitation wavelengths of 320 nm. The Ag-TPA CPNs are designed to detect ALP activity because silver ions can be reduced to silver nanoparticles. Owing to the enzymatic hydrolysis of AA2P into the reducing reagent AA in the presence of ALP, a novel fluorescent “turn-off” strategy for ALP activity assays using Ag-TPA CPNs is developed in this work (Figure 1).

## 2. Results and Discussion

### 2.1. Synthesis and Characterization of Ag-TPA CPNs

In Figure 1, the TEM images show that the shapes of Ag-TPA CPNs are irregular and the diameter range is 15–20 nm, as calculated from the 20 particle sizes in the TEM images using the “Nano Measurer 1.2” software. After adding ALP and its substrate AA2P, many silver nanoparticles with larger diameters of nearly 70–100 nm were formed owing to the reducing property of AA. In Figure 2A, the EDS elemental analysis shows that the atomic weight percentages of Ag, C, and O in the synthetic fluorescent Ag-TPA CPNs were 53.79%, 28.21%, and 18.01%, respectively. As shown in Figure 2B, the elemental mapping of C (a), O (b), and Ag (c) confirms the uniform distribution of the mentioned elements on Ag-TPA CPNs. All of these results clearly illustrate that the Ag-TPA CPNs were prepared successfully. As shown in Figure 3, the Ag-TPA CPN solution exhibited green fluorescent emissions under a 302 nm UV lamp. More importantly, it had two emission peaks at 490 nm and 520 nm under maximal excitation at 320 nm, with a large Stokes shift (~200 nm), which is preferable for the application of Ag-TPA CPNs in fluorescent bioanalyses. Moreover, the synthesized fluorescent Ag-TPA CPNs show high photostability. The Ag-TPA CPN solution can be made stable within two months by monitoring the fluorescent intensity, which is decreased to within 10% within two months. The Ag-TPA CPN powder that is put in the desiccators can be utilized for at least six months. The Ag-TPA CPNs exhibited a high fluorescent quantum yield (QY) of 20.19%, which was the absolute QY measured using an FLS 1000 photoluminescence spectrometer with an excitation wavelength of 320 nm.

### 2.2. The Feasibility of Fluorescent “Turn-Off” Detection of ALP Activity Using Ag-TPA CPNs

As shown in Figure 4, single ALP or AA2P components had little effect on the fluorescent intensity of the Ag-TPA CPNs. However, after adding AA or ALP to its substrate AA2P, the fluorescent intensity of Ag-TPA CPNs was significantly quenched. The fluorescent intensity of the Ag-TPA CPNs was quenched by 69.8% (fluorescent quenching efficiency) by AA2P with ALP and 84.4% by AA. The fluorescent quenching efficiency was calculated by (*I*_0_ − *I*)/*I*_0_, in which *I*_0_ represents the fluorescent intensity at 490 nm for Ag-TPA CPNs and *I* represents the fluorescent intensity at 490 nm for Ag-TPA CPNs after the addition of AA or ALP with the AA2P mixture. In addition, the solution showed a color change from colorless to dark-yellow and the fluorescent intensity was significantly decreased under the 302 nm UV lamp. According to previous studies, this color change is due to the localized surface plasmon resonance (LSPR) absorption of silver nanoparticles, which is reduced from silver ions by the product AA [34]. This fluorescent quenching signal of Ag-TPA CPNs is caused by the generation of AA, which is formed by the ALP-catalyzed hydrolysis of ascorbic acid 2-phosphate (AA2P) to ascorbic acid (AA), and the AA product triggers the reduction of Ag^+^ ions into silver nanoparticles. Therefore, this demonstrates the feasibility of the proposed fluorescent “turn-off” strategy for ALP activity detection.

### 2.3. The Possible Mechanism of Fluorescent “Turn-Off” Detection of ALP Activity

To further investigate the possible mechanism of the fluorescent “turn-off” determination of ALP activity using Ag-TPA CPNs, the UV–Vis absorption spectra and fluorescent lifetimes of different solutions were researched. As shown in Figure 5, the solutions of Ag-TPA, Ag-TPA-AA2P, Ag-TPA-ALP, AA, and AA2P-ALP had no distinct absorption peaks. However, after AA or AA2P-ALP was added to the Ag-TPA CPNs, within nearly 30 min at room temperature (25 °C), the corresponding solution showed an obvious UV absorption peak at 425 nm. ALP can trigger the catalyzed hydrolysis of AA2P into two products, AA and PO_4_^3−^, under alkaline conditions. AA is well-known as a reducing agent and is widely utilized in the synthesis of metal and metal oxide nanoparticles [35]. Therefore, silver ions in fluorescent Ag-TPA CPNs undergo a reducing process in the presence of AA or AA2P-ALP. Silver ions can be reduced to silver nanoparticles by AA, and the absorbance peak at 425 nm is ascribed to the LSPR absorption peak of silver nanoparticles [34,36,37]. This suggests that this fluorescent “turn-off” mechanism is related to the AA-triggered reduction of Ag^+^ ions into silver nanoparticles. We also measured the fluorescence of Ag-TPA CPNs in the presence of other metal ions such as Al^3+^, Ca^2+^, Zn^2+^, Cu^2+^, Fe^2+^, Pb^2+^, Mg^2+^, Ce^3+^, Co^2+^, and Eu^3+^, which could not make the fluorescence of the Ag-TPA CPNs “turn-off” (Appendix A). However, other reducing agents such as cysteine and glutathione could make the fluorescence of Ag-TPA CPNs “turn-off”, which helps to confirm the reduction mechanism of Ag^+^ ions into silver nanoparticles. However, low concentrations of cysteine and glutathione (5 μM or 10 μM) cannot make the fluorescence of Ag-TPA CPNs “turn-off” (Appendix A). Therefore, in diagnosing some diseases that have high concentrations of ALP but low concentrations of cysteine and glutathione, the presence of cysteine and glutathione may have little effect in detecting ALP. More importantly, since the detection target was ALP and the substrate of the enzyme ALP was AA2P, the proposed method was utilized in the fluorescent “turn-off” detection of ALP activity. During ALP detection, the concentration of AA2P (substrate) should be relatively high. When it is very low (5 μM or 10 μM), it cannot bring about effective fluorescent “turn-off” results.

The fluorescent “turn-off” mechanism was also further explored based on the fluorescent lifetimes of Ag-TPA CPNs in the absence and presence of ALP. As shown in Figure 6, the fluorescent lifetime of the Ag-TPA CPNs decreased from 3.93 ms to 3.80 ms after the addition of ALP. This suggests that this fluorescent “turn-off” detection of ALP activity is a dynamic quenching process instead of a static quenching process.

### 2.4. Optimization of the Experimental Conditions

Since the fluorescent intensity of Ag-TPA CPNs is affected by the time and temperature of the microwave reaction, the time and temperature of the microwave reaction were first optimized. As shown in Figure 7A, the fluorescent intensity of Ag-TPA CPNs increased with the increases in temperature from 25 °C to 80 °C, reaching the maximum value at 80 °C and then decreasing after 80 °C. Under high temperatures, Ag-TPA CPNs can become dark and may form silver oxide or hydroxide. In order to obtain the highest fluorescent intensity, 80 °C was chosen as the optimized microwave reaction temperature to synthesize fluorescent Ag-TPA CPNs.

As shown in Figure 7B, the fluorescent intensity of the Ag-TPA CPNs increased from 2 min to 15 min. Longer times can make Ag-TPA CPN solutions become dark and form silver oxide or hydroxide. Thus, 15 min was chosen as the optimized time to synthesize fluorescent Ag-TPA CPNs in order to obtain the maximum fluorescent intensity.

The effect of the pH on the fluorescent intensity of Ag-TPA CPNs was optimized using a HEPES buffer solution in the pH range of 5.0 to 10.5. As shown in Figure 7C, the fluorescent intensity of the Ag-TPA CPNs increased from pH 5.0 to 7.0 and then decreased significantly from 7.5 to 10.5. Under relatively high alkaline conditions, Ag-TPA CPNs may form silver oxide or hydroxide and show decreased fluorescent intensity. Thus, in order to obtain the maximum fluorescent intensity, 20 mM of HEPES buffer at pH 7.0 was chosen for the optimized conditions for the following fluorescent detection process.

In order to ensure the best fluorescent quenching efficiency of the Ag-TPA CPNs for ALP activity detection, the concentration of the AA2P and the incubation time and temperature of the ALP with AA2P were also investigated. As shown in Figure 7D, the fluorescent intensity of the Ag-TPA CPNs decreased gradually with the increasing concentrations of AA2P. Since the content of AA2P was low, the product of AA was also low at the same concentration of ALP. When the concentration of AA2P was higher than 0.25 mM, ALP reaches its maximum catalytic activity and no more AA was involved. Thus, the maximum quenching effect of the fluorescent intensity was obtained at 0.25 mM AA2P.

Similarly, the effect of the incubation temperature (25 °C, 37 °C, or 50 °C) of ALP with AA2P on the fluorescent quenching efficiency of the Ag-TPA CPNs was investigated (Figure 7E). High incubation temperatures may inactivate the enzyme activity of ALP. In order to achieve the maximum fluorescent quenching effect, 37 °C was chosen as the optimized incubation temperature for the following fluorescent detection process.

As shown in Figure 7F, the fluorescent intensity of the Ag-TPA CPNs was kept stable within 60 min. The fluorescent intensity of the Ag-TPA CPNs increased in the presence of ALP owing to fast ALP-catalyzed hydrolysis of AA2P to AA. Long reactions times favor the formation of more AA. When ALP hydrolyzes AA2P thoroughly and reaches the maximum catalytic activity, longer times do not lead to more AA and fluorescent quenching. Thus, to reach stability and the highest fluorescent quenching effect, 50 min was chosen as the optimized incubation time for the following fluorescent detection process.

### 2.5. The Performance of Fluorescent “Turn-Off” Detection of ALP Activity

Under the optimized conditions mentioned above, different concentrations of ALP were investigated for the fluorescent “turn-off” detection of ALP activity, as shown in Figure 8. Due to the gradual hydrolysis of AA2P and the increasing content of the AA product, the fluorescent intensity levels at 490 nm and 520 nm decreased gradually with the increasing concentrations of ALP from 0.2 to 12 mU/mL, reaching a plateau after 12 mU/mL (Figure 8A). The linear fitting equation was *I*_490 nm_ = 6.35–0.40 *c*_ALP_ (mU/mL), with a correlation coefficient (r) of 0.991 (Figure 8B). The limit of detection (LOD) for ALP was calculated to be 0.07 mU/mL using the formula of LOD = 3σ/k, where σ is the standard deviation of the blank solutions and k is the absolute value of the slope between the fluorescent intensity and the concentration of ALP. Moreover, compared with other reported methods (Table 1), this method using fluorescent Ag-TPA CPNs for ALP activity detection exhibits high sensitivity, a low LOD of 0.07 mU/mL, and a broad linear range of 0.2 to 12 mU/mL.

### 2.6. The Selectivity of the Fluorescent “Turn-Off” Detection of ALP Activity

As shown in Figure 9, only ALP causes a significant fluorescent quenching effect on the fluorescent intensity. Other metal ions (such as Al^3+^, Ca^2+^, Zn^2+^, Cu^2+^, Fe^2+^, Pb^2+^, and Mg^2+^), amino acids (such as glutamic acid, proline, alanine, and glycine), or enzymes (such as horseradish peroxidase (HRP), glucose oxidase (GOx), tyrosinase (Tyr), trypsin (Try), lysozyme (Lys), and superoxide (SOD)) show no obvious fluorescent changes. This indicates that this fluorescent “turn-off” method has good selectivity in ALP activity detection towards other metal ions, amino acids, or enzymes, which is beneficial for the fluorescent determination of ALP in biological samples.

### 2.7. Fluorescent “Turn-Off” Detection of ALP Activity in Serum Samples

In order to assess the feasibility of the proposed method for ALP detection in real serum samples, 10-fold-diluted commercial goat serum samples were utilized to monitor the ALP level, because human serum is difficult to obtain owing to the requirements of ethics and human privacy. Using the proposed method, no ALP was detected in the commercial goat serum sample. These commercial goat serum samples contained similar micromolar levels of cysteine, glutathione, and AA as real normal goat serum. However, they had a negligible fluorescent quenching effect in ALP detection. More importantly, commercial goat serum is 10-fold diluted, which means the concentration of the coexisting cysteine, glutathione, and AA are much lower. Therefore, as shown in Table 2, three diluted goat serum samples containing 1 mU/mL, 5 mU/mL, and 10 mU/mL of ALP were examined in the recovery experiments. The mean recovery levels ranged from 99.5% to 101.2%, with relative standard deviations (RSDs) lower than 4%. Even for the diluted goat serum samples with 10 μM cysteine, glutathione, and AA coexisting, recovery levels ranging from 97.5% to 100.7% were obtained. The satisfactory recovery levels and low RSD values indicate high accuracy of the proposed fluorescent “turn-off” method for the detection of ALP activity. This illustrates that this method is promising in the practical field of ALP analysis.

## 3. Conclusions

In summary, fluorescent Ag-TPA CPNs with a high quantum yield (20.19%) and large Stokes shift (~200 nm) were synthesized using a combination of TPA and silver nitrate using a microwave-assisted strategy within 15 min. This microwave-assisted synthesis method is rapid, economical, and highly efficient, providing a novel method for the synthesis of fluorescent Ag-TPA CPNs with excellent optical properties. Furthermore, the synthesized fluorescent Ag-TPA CPNs were applied in the fluorescent “turn-off” detection of ALP activity on the basis of the ALP-catalyzed hydrolysis of ascorbic acid 2-phosphate (AA2P) to ascorbic acid (AA), and the AA product triggered the reduction of Ag^+^ ions into silver nanoparticles. This fluorescent quenching detection method for ALP activity is a dynamic quenching process. It had high sensitivity for ALP activity detection, with a linear range of 0.2 to 12 mU/mL, a low LOD of 0.07 mU/mL, and high selectivity towards other metal ions, amino acids, and enzymes. Therefore, fluorescent Ag-TPA CPNs have great potential for application in ALP activity detection in biological systems.

## 4. Materials and Methods

### 4.1. Materials and Reagents

The silver nitrate (AgNO_3_) was purchased from Sinopharm Chemical Reagent Co. (Shanghai, China). The terephthalic acid (TPA) was obtained from Adamas-beta Reagent Co., Ltd. (Shanghai, China). The 2-phospho-L-ascorbic acid trisodium salt (AA2P), alkaline phosphatase (ALP), ascorbic acid (AA), 4-hydroxyethyl piperazine ethyl sulfonic acid (HEPES), tyrosinase (Tyr), horseradish peroxidase (HRP), glutamic acid (Glu), proline (Pro), alanine (Ala), and glycine (Gly) were purchased from Aladdin (Shanghai, China). The glucose oxidase (GOx), lysozyme (Lys), and trypsin (Try) were obtained from Shanghai Yuanye Bio-Technology Co., Ltd. (Shanghai, China). The superoxide dismutase (SOD) was obtained from Xiya Chemical Technology Co., Ltd. (Linyi, China). The N, N-dimethylformamide (DMF) was purchased from Kelon Chemical Reagent Factory (Chengdu, China). HEPES buffer solutions were used throughout all the experiments. All chemicals used in this work were of analytical grade and directly used without further purification.

### 4.2. Instrumentation

The fluorescent Ag-TPA CPNs were prepared using a MARS 6 CLASSIC microwave digestion workstation (Initiator 8 EXP, 2455 MHz frequency, 800 W) (American Analyx Corp., Boston, MA, USA). All fluorescent measurements were performed using a F98 fluorescence spectrometer (Lengguang Technology Ltd., Shanghai, China). An FLS 1000 photoluminescence spectrometer (Edinburgh Instruments Ltd., Edinburgh, England) was used to obtain the fluorescent lifetime of the fluorescent Ag-TPA CPNs. The UV–Vis absorption spectra were captured on a UV2550 spectrophotometer (Shimadzu Corporation, Kyushu, Japan). The transmission electron microscopy (TEM) images were obtained using a JEM-1200EX transmission electron microscope (JEOL Corp., Tokyo, Japan). An energy-dispersive spectrometry (EDS) analysis was performed using a Regulus 8010 scanning electron microscope (Hitachi Corp., Tokyo, Japan). A Youke pH S-3C digital pH meter (Shanghai Youke Instrument Co., Ltd, Shanghai, China) was used to measure the pH values of the aqueous solutions.

### 4.3. Synthesis of Fluorescent Ag-TPA CPNs

The fluorescent Ag-TPA CPNs were synthesized using a microwave-assisted process. Briefly, aqueous solutions of 5 mL of 0.05 M AgNO_3_ and 5 mL of 0.05 M terephthalic acid (TPA), which was dissolved using N, N-dimethylformamide (DMF), were added into a 50 mL centrifuge tube. The whole solution was mixed for 1 min at room temperature using a vortex mixer. Then, the mixture was placed together in a 30 mL microwave tube, heated by a microwave (800 W) under autogenous pressure at 80 °C for 15 min, and then cooled down naturally to room temperature. The white suspension was centrifuged at 8000 rpm and washed with ethanol and ultrapure water three times to remove redundant ligands and metal ions. Finally, the product was collected and redispersed into 10 mL of water. In the following ALP detection process, 50 μL Ag-TPA CPN solutions were used in each measurement.

### 4.4. Fluorescent “Turn-Off” Detection of ALP Activity Using Ag-TPA CPNs

For the measurement of ALP activity, 50 μL of 5 mM AA2P firstly reacted with different concentrations of ALP in 50 μL of 20 mM HEPES buffer (pH 9.0) at 37 °C for 50 min. Then, 300 μL of 20 mM HEPES buffer (pH 7.0) and 50 μL of Ag-TPA CPN solution were added. An appropriate amount of water was added to keep the final volume of the whole solutions at 1 mL. After thorough vortex mixing and reaction at 25 °C for 30 min, the fluorescent intensities were recorded on a fluorescent spectrophotometer with an excitation wavelength at 320 nm. The slit widths for excitation and emission were 10 nm and 10 nm, respectively. The photomultiplier tube voltage (PMT) was kept at 800 V.

### 4.5. Fluorescent “Turn-Off” Detection of ALP Activity in Real Goat Serum Samples

The standard additive method was used to detect ALP activity in goat serum samples. Since no ALP was detected in the goat serum samples, 1 mU/mL, 5 mU/mL, and 10 U/mL samples of ALP were respectively spiked into the goat serum samples. The other fluorescent detection processes were similar to the ALP activity detection method mentioned above.

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
