# Peer review of "Ultrarapid Microwave-Assisted Synthesis of Fluorescent Silver Coordination Polymer Nanoparticles and Its Application in Detecting Alkaline Phosphatase Activity"

_molecules, 2023, doi:10.3390/molecules28041892_

Round 1

Reviewer 1 Report

This work described a rapid microwave-assisted synthesis of fluorescent silver coordination polymer nanoparticles (Ag-TPA CPNs) and applied them in fluorescent quenching detection of ALP activity in the range of 0.2-12 mU/mL. It shows high sensitivity and selectivity in ALP activity detection. All the experimental results and sufficient detail are adequately documented, which is original and scientific reliable. However, there are some issues that the authors should pay attention to. Therefore minor revisions before the acceptance for publication are suggested.

1. The limit of detection (LOD) in Section 2.5 “The performance of fluorescence “turn-off” detection of ALP activity” and Abstracts is inconsistent. Please check it.

2. Some recent references about ALP detection or fluorescent nanoparticles can be cited.

3. There are few minor mistakes including spelling errors and grammatical errors. Please check them carefully throughout the manuscript.

Author Response

Response to the comments

Manuscript No.: molecules-2181851

Title: Ultrarapid Microwave-assisted Synthesis of Fluorescent Silver Coordination Polymer Nanoparticles and Its Application in Detecting Alkaline Phosphatase Activity

Journal: Molecules

Corresponding author: Prof. Wenjing Qi

Dear Editor,

Enclosed is a revised manuscript (molecules-2181851). We are very grateful for the constructive comments. We have carefully revised our manuscript, and highlighted the changes to the manuscript. The point-by-point response to the comments of the reviewers is listed in the file named as “Responses to reviewers”.

Response to Reviewer #1's comments:

This work described a rapid microwave-assisted synthesis of fluorescent silver coordination polymer nanoparticles (Ag-TPA CPNs) and applied them in fluorescent quenching detection of ALP activity in the range of 0.2-12 mU/mL. It shows high sensitivity and selectivity in ALP activity detection. All the experimental results and sufficient detail are adequately documented, which is original and scientific reliable. However, there are some issues that the authors should pay attention to. Therefore minor revisions before the acceptance for publication are suggested.

  1. The limit of detection (LOD) in Section 2.5 “The performance of fluorescence “turn-off” detection of ALP activity” and Abstracts is inconsistent. Please check it.

Thank you for your helpful comments. We have checked it and used “limit of detection (LOD)” in Abstracts (Page 1) and the whole revised manuscript.

  1. Some recent references about ALP detection or fluorescent nanoparticles can be cited.

Thank you for your helpful comments. We have added recent references about ALP detection or fluorescent nanoparticles. Please see “References” in Page 14 [Anal. Chem. 2023, doi: 10.1021/acs.analchem.1022c04327; Molecules. 2022, 28, 181; Molecules. 2022, 27, 6440].

  1. There are few minor mistakes including spelling errors and grammatical errors. Please check them carefully throughout the manuscript?

Thank you for your helpful comments. According to the comments from three reviewers, we have checked spelling and grammatical problems carefully throughout the manuscript.

Reviewer 2 Report

This work described a rapid microwave-assisted synthesis of fluorescent Ag-TPA CPNs and utilized them in fluorescent quenching detection of ALP from 0.2 mU/mL to 12 mU/mL. It shows high sensitivity with LOD of 0.07 mU/mL and high selectivity towards metal ions, amino acids and other enzymes such as HRP, GOx, tyrosinase, trypsin, lysozyme or superoxide. All the experimental results and sufficient detail are adequately documented. But I suggest minor revisions before the acceptance for publication because there are some issues that the authors should revise. 

1. How long can fluorescent Ag-TPA CPNs be stable? The stability of Ag-TPA CPNs can be demonstrated.

2. Please check the demonstration. Is “fluorescent “turn-off” detection” or “fluorescence “turn-off” detection” OK? It should be the same in the whole mansucript.

3. Some recent references about fluorescent detection or other ALP detections can be added.

4. Some minor mistakes need to be carefully checked throughout the manuscript. For example, spelling errors and grammatical errors. 

Author Response

Response to the comments

Manuscript No.: molecules-2181851

Title: Ultrarapid Microwave-assisted Synthesis of Fluorescent Silver Coordination Polymer Nanoparticles and Its Application in Detecting Alkaline Phosphatase Activity

Journal: Molecules

Corresponding author: Prof. Wenjing Qi

Dear Editor,

Enclosed is a revised manuscript (molecules-2181851). We are very grateful for the constructive comments. We have carefully revised our manuscript, and highlighted the changes to the manuscript. 

Response to Reviewer #2's comments:

This work described a rapid microwave-assisted synthesis of fluorescent Ag-TPA CPNs and utilized them in fluorescent quenching detection of ALP from 0.2 mU/mL to 12 mU/mL. It shows high sensitivity with LOD of 0.07 mU/mL and high selectivity towards metal ions, amino acids and other enzymes such as HRP, GOx, tyrosinase, trypsin, lysozyme or superoxide. All the experimental results and sufficient detail are adequately documented. But I suggest minor revisions before the acceptance for publication because there are some issues that the authors should revise.
1. How long can fluorescent Ag-TPA CPNs be stable? The stability of Ag-TPA CPNs can be demonstrated.

Thank you for your helpful comments. We have added it. Please see Page 3.

“Moreover, the synthesized fluorescent Ag-TPA CPNs shows high photostability. Ag-TPA CPNs solution can be stable within two months by monitoring fluorescent intensity, which gets dereased wihtin 10 % within two months. Ag-TPA CPNs powder which is put in desiccators can be utilized at least six months.”

  1. Please check the demonstration. Is “fluorescent “turn-off” detection” or “fluorescence “turn-off” detection” OK? It should be the same in the whole manuscript.

Thank you for your helpful comments. We have checked it. According to various references, both “fluorescent “turn-off” detection” and “fluorescence “turn-off” detection” are OK. We have revised it and used “fluorescent “turn-off” detection” in the whole revised manuscript.

  1. Some recent references about fluorescent detection or other ALP detections can be added.

Thank you for your helpful comments. We have added recent references recent references about fluorescent detection or other ALP detections. Please see “References” [Anal. Chem. 2023, doi: 10.1021/acs.analchem.1022c04327; Molecules. 2022, 28, 181; Molecules. 2022, 27, 6440] in Page 14.

  1. Some minor mistakes need to be carefully checked throughout the manuscript. For example, spelling errors and grammatical errors.

Thank you for your helpful comments. According to the comments from three reviewers, we have checked spelling and grammatical problems carefully throughout the manuscript.

Reviewer 3 Report

In my opinion, the manuscript should be modified carefully before being published.

1. The figure caption of Fig.1, scale bars: 400 nm. But in the TEM images, the scale bars are not.

2. Line 98: the diameter is 15-20 nm, but in the TEM image, the scale bar is 500 nm. It is not a proper TEM image. I think all the TEM images in Fig 1 should be redone.

3. Line 124: the process of calculating the quenched percentages should be described in detail.

4. As described in the introduction part, the ALP detection is very important for several diseases, why goat serum samples were selected, instead of human serum?

5. How the QY was obtained? Is it absolute QY or relative QY?

6. There are two emission peaks, which one was selected for the calculations in the manuscript?

7. For the selectivity of this method, some metal ions, enzymes and amino acids were applied.

Because the method was established based on the reduction of AA, how about the substances with oxidability or reducibility?

8. In the optimization of experimental conditions, the results were listed, but why these results were obtained, what are the explanations for the results?

9. The concentrations of reactants should be shown in the experiment, such as Fig 3-5 and Fig7, otherwise the experiment cannot be repeated.

10. In table 1, the “detection mechanism” is not suitable to list “on-off” or “off-on”.

Author Response

Response to the comments

Manuscript No.: molecules-2181851

Title: Ultrarapid Microwave-assisted Synthesis of Fluorescent Silver Coordination Polymer Nanoparticles and Its Application in Detecting Alkaline Phosphatase Activity

Journal: Molecules

Corresponding author: Prof. Wenjing Qi

Dear Editor,

Enclosed is a revised manuscript (molecules-2181851). We are very grateful for the constructive comments. We have carefully revised our manuscript, and highlighted the changes to the manuscript. The point-by-point response to the comments of the reviewers is listed in the file named as “Responses to reviewers”.

Response to Reviewer #3's comments:

This work described a rapid microwave-assisted synthesis of fluorescent silver coordination polymer nanoparticles (Ag-TPA CPNs) and applied them in fluorescent quenching detection of ALP activity in the range of 0.2-12 mU/mL. It shows high sensitivity and selectivity in ALP activity detection. All the experimental results and sufficient detail are adequately documented, which is original and scientific reliable. However, there are some issues that the authors should pay attention to. Therefore minor revisions before the acceptance for publication are suggested.

  1. The figure caption of Fig.1, scale bars: 400 nm. But in the TEM images, the scale bars are not.
  2. Line 98: the diameter is 15-20 nm, but in the TEM image, the scale bar is 500 nm. It is not a proper TEM image. I think all the TEM images in Fig 1 should be redone.

Thank you for your helpful comments. According to the comments from three reviewers, we have revised TEM images. Please see figure in Page 3-4.

“As shown in Figure 1, TEM image shows that the shapes of Ag-TPA CPNs are irregular particles and the diameter is 15-20 nm, which are calculated from 20 particle size in TEM mages using “Nano Measurer 1.2” software.”

“Moreover, the synthesized fluorescent Ag-TPA CPNs shows high photostability. Ag-TPA CPNs solution can be stable within two months by monitoring fluorescent intensity, which gets decreased within 10 % within two months. Ag-TPA CPNs powder which is put in desiccators can be utilized at least six months. Ag-TPA CPNs exhibit high fluorescent quantum yield (QY) of 20.19%, which is a absolute QY measured by FLS 1000 photoluminescence spectrometer with the excitation wavelength of 320 nm”.

Figure 1. TEM image of Ag-TPA CPNs in the absence (A,C) and presence (B, D) of ALP.

  1. Line 124: the process of calculating the quenched percentages should be described in detail.

Thank you for your helpful comments. According to your comments, we have added the demonstrations. Please see Page 5 Section 2.2 “The feasibility of fluorescent “turn-off” detection of ALP activity using Ag-TPA CPNs”.

“As shown in Figure 4, single ALP or AA2P has little effect on fluorescent intensity of Ag-TPA CPNs. However, after adding AA or ALP with its substrates AA2P, fluorescent intensity of Ag-TPA CPNs is significantly quenched. Fluorescent intensity of Ag-TPA CPNs gets quenched 69.8% (fluorescent quenching efficiency) by AA2P with ALP and 84.4% by AA. Fluorescent quenching efficiency is calculated by (I0 - I)/I0, in which I0 represents fluorescent intensity of Ag-TPA CPNs; I represents Ag-TPA CPNs after the addition of ALP with AA2P mixture or AA”.

  1. As described in the introduction part, the ALP detection is very important for several diseases, why goat serum samples were selected, instead of human serum?

Thank you for your helpful comments. Owing to the requirements of ethics and human privacy, human serum is difficult to obtain. Many reported references utilized commercial serum samples in bioanalysis.

“In order to assess the feasibility of the proposed method for ALP detection in real serum samples, 10-fold diluted commercial goat serum samples are utilized to monitor ALP level, because human serum is difficult to obtain owing to the requirements of ethics”. Please see Page 12.

  1. How the QY was obtained? Is it absolute QY or relative QY?

Thank you for your helpful comments. We have added more details. Please see Page 3.

“Ag-TPA CPNs exhibit high fluorescent quantum yield (QY) of 20.19%, which is an absolute QY measured by FLS 1000 photoluminescence spectrometer with the excitation wavelength of 320 nm.”

  1. There are two emission peaks, which one was selected for the calculations in the manuscript?

Thank you for your helpful comments. We have added more details. Please see Figure 7-9. lex: 320 nm; lem: 490 nm

  1. For the selectivity of this method, some metal ions, enzymes and amino acids were applied.

Because the method was established based on the reduction of AA, how about the substances with oxidability or reducibility?

Thank you for your helpful comments. We have done the experiment before and put the data in supporting information. Please see Page 6 and figure S1 (supporting information).

“We also did the fluorescence of Ag-TPA CPNs in the presence of other metal ions such as Al3+, Ca2+, Zn2+, Cu2+, Fe2+, Pb2+, Mg2+, Ce3+, Co2+ and Eu3+. They cannot make fluorescence of Ag-TPA CPNs “turn-off” (Figure S1 in supporting information). While other reducing agents such as cysteine and glutathione can also make fluorescence of Ag-TPA CPNs “turn-off”, which helps to confirm reduction mechanism of Ag+ ions into silver nanoparticles. Since the detection target is ALP and the substrate of enzyme ALP is AA2P, the proposed method is utilized in fluorescent “turn-off” detection of ALP activity”.

Figure S1. Fluorescent intensity of Ag-TPA CPNs in the presence of 100 M AA and other species (Al3+, Ca2+, Zn2+, Cu2+, Fe2+, Pb2+, Mg2+, Ce3+, Co2+, Eu3+, Cys, GSH). Cys: cysteine; GSH: glutathione. Ag-TPA CPNs solution: 50 mL; lex: 320 nm; lem: 490 nm. All the error bars represent the standard deviation of three measurements.

  1. In the optimization of experimental conditions, the results were listed, but why these results were obtained, what are the explanations for the results?

Thank you for your helpful comments. We have added more demonstrations. Please see Section 2.4 “The optimization of experimental conditions” in Page 7-9.

“Since fluorescent intensity of Ag-TPA CPNs is affected by the time and temperature of microwave reaction, the time and temperature of microwave reaction are firstly optimized. As shown in Figure 7A, fluorescent intensity of Ag-TPA CPNs gets increased with the increasing temperature from 25 oC to 80 oC, reach the maximum value at 80 oC and then get decreased after 80 oC. Under high temperature, Ag-TPA CPNs can become dark and may form silver oxide or hydroxide. In order to get the higheset fluorescent intensity, 80 oC is chosen as the optimized microwave reaction conditions to synthesize fluorescent Ag-TPA CPNs.

As shown in Figure 7B, fluorescent intensity of Ag-TPA CPNs gets increased from 2 min to 15 min. Longer time can make Ag-TPA CPNs solutions become dark and may form silver oxide or hydroxide. Thus, 15 minutes is chosen as the optimized conditions to synthesize fluorescent Ag-TPA CPNs in order to obtain the maximum fluorescent intensity.

The effect of pH on fluorescent intensity of Ag-TPA CPNs is optimized using HEPES buffer solution at pH range from 5.0 to 10.5. As shown in Figure 7C, fluorescent intensity of Ag-TPA CPNs gets increased from pH 5.0 to 7.0 and then decreased significantly from 7.5 to 10.5. Under relatively high alkaline conditions, Ag-TPA CPNs may form silver oxide or hydroxide and decrease fluorescent intensity. Thus in order to obtain the maximum fluorescent intensity, 20 mM HEPES buffer pH 7.0 is chosen as the optimized conditions for the following fluorescent detection.

In order to ensure the best fluorescent quenching efficiency of Ag-TPA CPNs for ALP activity detection, the concentrations of AA2P, incubation time and temperature of ALP with AA2P are also investigated. As shown in Figure 7D, fluorescent intensity of Ag-TPA CPNs gets decreased gradually with the increasing concentrations of AA2P. Since the content of AA2P is low, the product of AA is also low at the same concentration of ALP. When the concentration of AA2P is higher than 0.25 mM, ALP reaches its maximum catalytic activity and no more AA is brought. Thus the maximum quenching effect of fluorescent intensity is obtained at 0.25 mM AA2P.

Similarly, the effect of incubation temperature (25 oC, 37 oC and 50 oC) of ALP with AA2P on fluorescent quenching efficiency of Ag-TPA CPNs is investigated (Figure 7E). High incubation temperature may make the enzyme activity of ALP inactivate. In order to get the maximum fluorescent quenching effect, 37 oC is chosen as the optimized incubation temperature for the following fluorescent detection.

As shown in Figure 7F, fluorescent intensity of Ag-TPA CPNs is kept stable within 60 min. Fluorescent intensity of Ag-TPA CPNs gets increased in the presence of ALP owing to fast ALP-catalyzed hydrolysis of AA2P to AA. Long time is in favor of the formation of more AA. When ALP hydrolyzes AA2P thoroughly and reaches the maximum catalytic activity, longer time cannot bring more AA and fluorescent quenching. Thus to reach stable and the highest fluorescent quenching effect, 50 min is chosen as the optimized incubation time for the following fluorescent detection”.

Figure 7. The effect of different microwave irradiation temperature (25 oC, 40 oC, 60 oC, 80 oC, 100 oC, 120 oC) (A), time (2 – 30 min) (B) and pH (5.0 – 10.5) (C) on fluorescent intensity of Ag-TPA CPNs. The effect of different concentrations of AA2P (0.05 - 0.40 mM) (D), reaction temperature (25 oC, 37 oC, 50 oC) (E) and time (10 – 60 min) (F) on fluorescent detection of ALP. c(ALP): 10 mU/mL; c(AA2P): 0.25 mM; Ag-TPA CPNs solution: 50 mL; lex: 320 nm; lem: 490 nm; All the error bars represent the standard deviation of three measurements.

  1. The concentrations of reactants should be shown in the experiment, such as Fig 3-5 and Fig7, otherwise the experiment cannot be repeated.

Thank you for your helpful comments. We have added the concentrations of reactants and other experimental details in the revised manuscript. Please see figure 3-9.

  1. In table 1, the “detection mechanism” is not suitable to list “on-off” or “off-on”.

Thank you for your helpful comments. We have revised it. “Detection signal change” is more suitable than “detection mechanism”. Please see Table 1 in Page 10-11.

Table 1 The comparison of different methods for the detection of ALP activity.

Thanks again for your helpful review!

Round 2

Reviewer 3 Report

In the revised manuscript and the Supporting Information, the reducing agents such as cysteine, glutathione and ascorbic acid can also make fluorescence of Ag-TPA CPNs “turn-off” greatly, so, when using this method to detect real samples, the influences caused by these reducing agents cannot be ignored. Because they are very likely or certain to exist in serum. In my opinion, more experiments are needed to prove that, the existence of these agents has little effect to detect ALP in this method.

Author Response

Response to the comments

Manuscript No.: molecules-2181851

Title: Ultrarapid Microwave-assisted Synthesis of Fluorescent Silver Coordination Polymer Nanoparticles and Its Application in Detecting Alkaline Phosphatase Activity

Journal: Molecules

Corresponding author: Prof. Wenjing Qi

Dear Editor,

Enclosed is a revised manuscript (molecules-2181851). We are very grateful for the constructive comments. We have carefully revised our manuscript, and highlighted the changes to the manuscript. The point-by-point response to the comments of the reviewers is listed in the file named as “Responses to reviewers”.

Response to Reviewer #3's comments:

  1. In the revised manuscript and the Supporting Information, the reducing agents such as cysteine, glutathione and ascorbic acid can also make fluorescence of Ag-TPA CPNs “turn-off” greatly, so, when using this method to detect real samples, the influences caused by these reducing agents cannot be ignored. Because they are very likely or certain to exist in serum. In my opinion, more experiments are needed to prove that, the existence of these agents has little effect to detect ALP in this method.

Thank you for your helpful comments to improve my manuscript. We have added more experiments.

“We also did the fluorescence of Ag-TPA CPNs in the presence of other metal ions such as Al3+, Ca2+, Zn2+, Cu2+, Fe2+, Pb2+, Mg2+, Ce3+, Co2+ and Eu3+. They cannot make fluorescence of Ag-TPA CPNs “turn-off” (Figure S1 in supporting information). While other reducing agents such as cysteine and glutathione can also make fluorescence of Ag-TPA CPNs “turn-off”, which helps to confirm reduction mechanism of Ag+ ions into silver nanoparticles. However, low concentration of cysteine and glutathione (5 mM or 10 mM) cannot make fluorescence of Ag-TPA CPNs “turn-off” (Figure S1). Therefore in diagnosing some diseases which have high concentration of ALP but low concentration of cysteine and glutathione, the presence of cysteine and glutathione may have little effect to detect ALP. More importantly, since the detection target is ALP and the substrate of enzyme ALP is AA2P, the proposed method is utilized in fluorescent “turn-off” detection of ALP activity. During ALP detection, the concentration of AA2P (substrate) should be relative high. When it is very low (5 mM or 10 mM), it cannot bring effective fluorescent “turn-off” result.” Please see Page 6-7.

Please see Page 12 Section 2.7 “Fluorescent “turn-off” detection of ALP activity in serum samples. “In order to assess the feasibility of the proposed method for ALP detection in real serum samples, 10-fold diluted commercial goat serum samples are utilized to monitor ALP level, because human serum is difficult to obtain owing to the requirements of ethics and human privacy. Using the proposed method, no ALP is detected in commercial goat serum sample. These commercial goat serum samples contain similar micromolar level cysteine, glutathione or AA as real normal goat serum. But it has negligible fluorescent quenching effect in ALP detecting. More importantly, commercial goat serum is 10-fold diluted, which makes the concentration of coexisting cysteine, glutathione or AA be much lower. Therefore, as shown in Table 2, three diluted goat serum samples containing 1 mU/mL, 5 mU/mL and 10 mU/mL ALP are examined by the recovery experiments. Mean recoveries are ranged from 99.5% to 101.2% with relative standard deviations (RSDs) lower than 4%. Even if diluted goat serum samples with 10 mM cysteine, glutathione and AA coexisting, recoveries from 97.5% to 100.7% are obtained. Satisfactory recoveries and low RSD values indicate high accuracy of the proposed fluorescent “turn-off” method for detection of ALP activity. It illustrates that this method is promising in practical field of ALP analysis.”

  According to many reports, some diseases including liver dysfunction or breast cancer may have high level of ALP, cysteine and glutathione. It is indeed difficult to detect ALP in the patients’ serum. It is a fact and a difficult problem. But in some fields, the proposed method in this work is meaningful and practical. For instance, the normal range of serum ALP in adults is about 46 -190 U/L, while children and pregnant women almost have higher levels (more than 500 U/L) [Biosensors & Bioelectronics. 2013, 43: 366-371; Clinica Chimica Acta, 1967, 15: 241-245; Clin Invest Med, 2019, 42: E47-52]. But the concentration of cysteine, glutathione or AA is only micromolar level. The large concentration difference makes the possible coexisting cysteine, glutathione and AA have little effect on fluorescent detection of ALP. Besides, since the high level of ALP in children and pregnant women, the serum samples need to be diluted 10-20 fold before ALP detection because the proposed method in this work has high sensitivity for ALP activity detection with linear range from 0.2 to 12 mU/mL and low LOD of 0.07 mU/mL. When serum samples are 10-20 fold diluted, the possible level of cysteine, glutathione or AA is much low and it has little effect to detect ALP. More importantly, during ALP detection using the proposed method in this work, 0.25 mM AA2P is needed. When AA2P is low and the product AA should be low, no obvious fluorescent “turn-off” result can be seen.

Therefore, a novel developed ALP detection method may not be suitable for all detection fields, but in some fields it is indeed an effective method. It should be meaningful for the research.

Thank you for giving us the chance!

Thanks you for your kind understanding and helpful review!

Your consideration will be greatly appreciated!

Table 2. Recoveries of ALP activity detection in serum samples.

Samples

Added ALP

 (mU/mL)

Found ALP

(mU/mL, n = 3)

Recoveries

(%, n = 3)

1

1

1.01, 1.03, 0.98

100.7 ± 2.5

2

5

4.78, 5.13, 5.02

99.5 ± 3.6

3

10

9.88, 10.05, 10.43

101.2 ± 2.8

4 a

10

9.66, 9.82, 10.23

99.0 ± 2.9

5 b

10

9.69, 9.85, 9.70

97.5 ± 0.9

6 c

10

9.65, 10.42, 10.15

100.7 ± 3.9

a represents serum sample with 10 mM cysteine coexisting;

b represents serum sample with 10 mM glutathione coexisting;

c represents serum sample with 10 mM cysteine, glutathione and AA coexisting;

Figure S1. Fluorescent intensity of Ag-TPA CPNs in the presence of AA (100, 10 and 5 mM) and other species (Al3+, Ca2+, Zn2+, Cu2+, Fe2+, Pb2+, Mg2+, Ce3+, Co2+, Eu3+, Cys, GSH, 100 mM). Cys: cysteine; GSH: glutathione. Ag-TPA CPNs solution: 50 mL; lex: 320 nm; lem: 490 nm. All the error bars represent the standard deviation of three measurements.

Best wishes,

Wenjing Qi (W Qi)

Professor of Chemistry

Chongqing Normal University

Chongqing 401331, PR China

Tel: +86-23-65362777

E-mail: wenjingqi616@cqnu.edu.cn (Prof. W. Qi)
